# Effects of Incorporation of Essential Oils in the Jersey Cow Diet on the Quality of Produced Dairy Products (Milk, Cream, and Colonial Cheese)

**DOI:** 10.3390/foods14162788

**Published:** 2025-08-11

**Authors:** Cristina Bachmann da Silva, Aline Zampar, Beatriz Danieli, Aline Luiza do Nascimento, Lucas Henrique Bavaresco, Elisandra Rigo, Bruna Klein, Alline Artigiani Lima Tribst, Karen Karine da Rosa Dias, Fabiana Quoos Mayer, Ana Paula Muterle Varela, Michele Mann, Jeverson Frazzon, Creciana Maria Endres, Ana Luiza Bachmann Schogor

**Affiliations:** 1Graduate Program in Animal Science, Santa Catarina State University (UDESC), Chapecó 89815-630, SC, Brazil; cristinabachmann5@gmail.com (C.B.d.S.); alineluizan@outlook.com (A.L.d.N.); lucas.h.bavaresco@gmail.com (L.H.B.); 2Department of Animal Science, UDESC, Chapecó 89815-630, SC, Brazil; aline.zampar@udesc.br (A.Z.); beatriz.danieli@udesc.br (B.D.); 3Department of Food and Chemical Engineering, UDESC, Pinhalzinho 89870-000, SC, Brazil; elisandra.rigo@udesc.br (E.R.); brunaklein06@yahoo.com.br (B.K.); 4Center for Food Studies and Research (NEPA), State University of Campinas (UNICAMP), Campinas 13083-852, SP, Brazil; tribst@unicamp.br; 5Post Graduate Program in Cellular and Molecular Biology, Biotechnology Center, Federal University of Rio Grande do Sul—UFRGS, Porto Alegre 91501-970, RS, Brazil; karenkrdias@gmail.com (K.K.d.R.D.); bimmayer@gmail.com (F.Q.M.); 6Department of Food Science, Federal University of Rio Grande do Sul (UFRGS), Porto Alegre 90040-060, RS, Brazil; anapaulamut@gmail.com (A.P.M.V.); mbertonimann@gmail.com (M.M.); jevfrazzon@gmail.com (J.F.); 7SENAI/SC University Center—UniSENAI Campus Chapecó, Chapecó 89803-785, SC, Brazil

**Keywords:** dairy products, colonial cheese, fatty acid profile, sensory evaluation, microbiome, lipid oxidation

## Abstract

Essential oil blends (EOBs) have been increasingly studied for their multifunctional benefits in animal nutrition and food science. This study evaluates the impact of an EOB composed of eucalyptus and peppermint oil on the physicochemical, microbiological, and sensory characteristics of dairy products—milk, cream, and colonial cheese. Forty lactating cows were assigned to two groups: control and EOB addition (3.6 mL/cow/day) on the diet. Sensory analysis showed that the addition of EOB does not significantly alter milk or cream characteristics but enhances cheese texture perception. Fatty acid analysis revealed a higher proportion of SFA and lower UFA in cheese produced from the EOB group. Additionally, the EOB reduced lipid oxidation throughout the ripening process, with significantly lower TBARS values at 45 days of maturation (0.1300), compared to those from cows without supplementation of EOB (0.1585), suggesting improved oxidative stability. Microbiome analysis indicated that the bacterial community composition remained stable, with a slight reduction in *Streptococcus* spp. in EOB cheeses. No drastic shifts in microbial diversity were detected, and a lower overall abundance of bacterial taxa was observed in the EOB group. Results suggest that EOBs in dairy cow diets may positively modulate dairy product characteristics and alter the microbiota without compromising sensory quality. This study highlights the technological potential of EOB supplementation in dairy production.

## 1. Introduction

Essential oils (EOs) have gained increasing attention in animal nutrition and food science due to their multifunctional properties, including antimicrobial, antioxidant, and sensory-enhancing effects. In the dairy industry, the incorporation of EOs into milk and dairy products has demonstrated several benefits, particularly through the reduction of microbial activity in dairy matrices [1]. Furthermore, EOs have been reported to enhance antioxidant activity, improve organoleptic properties, increase stability, and prolong shelf life [1].

When EOs are administered through the diet of dairy cows, the resulting effects on milk and dairy products can differ. Recent studies have shown significant improvements in animal health and productivity when essential oil blends (EOBs) are included in the diet. For instance, EOBs composed of clove, oregano, and juniper have been linked to increased milk yield, improved feed efficiency, better milk composition, and reduced somatic cell count (SCC) [2]. Similarly, supplementation in heat-stressed ewes with oregano and thyme oils increased milk production and decreased SCC [3]. Cows supplemented with eucalyptus, thyme, and fennel oils demonstrated increased dry matter intake (DMI) and improved respiratory health, although without significant changes in milk yield or composition [4].

From a technological standpoint, EOB supplementation has also been associated with improved oxidative stability in milk and dairy products [5]. However, due to the oily nature of EOBs, there is a growing interest in evaluating their impact on the milk fatty acid profile. Research has indicated that blends containing oregano, cumin, cinnamon, and garlic oils can significantly modify this profile [6].

Despite these benefits, the effectiveness of EO supplementation can vary based on the active compounds used, their concentrations, and the method of administration. While it is known that EO levels in milk are lower than the ingested amounts, there is limited information regarding the sensory perception and consumer acceptability of EO-treated dairy products.

Menthol, a major compound in peppermint oil (*Mentha piperita* L.), is recognized for its bactericidal properties [7], antioxidant activity [8], and distinct aroma [9]. Cineole, found in eucalyptus oil (*Eucalyptus* spp.), exhibits antibiotic, anti-inflammatory, acaricidal, antifungal, and antioxidant effects [10]. The fatty acid compositions of these oils are well documented [11,12].

To date, EOBs combining eucalyptus oil, peppermint oil, and menthol crystals have shown promise in preventing respiratory diseases in calves [13] and piglets [14]. However, their application in dairy cows has not yet been explored. As such, their potential to improve animal health, alter milk composition, and affect the sensory characteristics of dairy products remains unknown.

It is hypothesized that supplementation of dairy cows with this EOB may alter the milk fatty acid profile and positively influence the microbiota of colonial cheese throughout its ripening process. Furthermore, it is believed that this additive will not be perceptible in sensory analyses of milk and dairy products, indicating its technological feasibility and consumer acceptability. Our objective was to characterize the physicochemical and microbiological composition of milk, cream, and colonial cheese produced by cows that received an EOB composed of eucalyptus oil, peppermint oil, and menthol crystals. In addition, we assessed the effects of this EOB on the fatty acid profile of colonial cheese at different ripening stages, as well as on the sensory detection of the EOB in milk, cream, and colonial cheese. Finally, we evaluated the micro-biome of colonial cheese at different ripening stages.

## 2. Materials and Methods

The study was conducted on a commercial farm located in Guatambu, Santa Catarina, Brazil. In May of 2023, forty healthy, multiparous, parasite-free Jersey cows (429 ± 40 kg body weight), at approximately 110 days in milk (DIM), producing ~33 kg of milk per day, were assigned to a completely randomized design. Cows were grouped into two treatment groups based on days in milk and milk yield and housed in a compost barn with internal divisions. All the procedures were approved by the Animal Research Ethics Committee of the State University of Santa Catarina, No. 5448251022.

An essential oil blend (BronchoVest, Biochem, Germany) was either incorporated into the total mixed ration (TMR) or omitted, resulting in two treatments: (1) TMR plus 3.6 mL/cow/day of essential oil blend (EOB; n = 20) and (2) TMR without essential oils (Control; n = 20). The dosage of the EOB provided followed the recommendations indicated on the commercial product label. The EOB group received 72 mL/day for the 20 cows, while the control group received an equivalent volume of placebo (distilled water).

The EOB consisted of three sources of two active compounds, cineole and menthol: eucalyptus oil (157.9 g/L), peppermint oil (32 g/L), and menthol crystals (55 g/L). The daily EOB dose (72 mL) was diluted in nine liters of distilled water prior to use, according to the manufacturer’s recommendations (Biochem do Brasil Nutrição Animal LTDA (Guarulhos, SP, Brazil), registered under MAPA No. 245/2020/UTVDA/DREP). The diluted EOB and placebo were homogenized into the TMR using a forage wagon prior to the morning and afternoon feedings.

The experimental period began at 110 DIM and consisted of a 14-day adaptation phase followed by 17 days of data and milk collection, totaling 31 days. Animals received EOB supplementation (EOB group) or placebo (control group) throughout the entire experimental period.

The cows were fed a TMR provided twice daily, at 6:00 a.m. and 5:00 p.m. The TMR was formulated to meet the nutritional requirements of Jersey cows at 110 DIM, with a body weight of 425 kg, producing 28.2 kg of milk per day, with 4.4% fat and 3.5% protein, according to NASEN (2021) guidelines. Approximately 10 kg of dry matter (DM) per dairy cow from the TMR was offered at each feeding. The ingredients of the TMR were (g/kg of DM basis): corn silage, 433; wheat silage, 122; commercial concentrate with 24% crude protein, 431, and commercial mineral and vitamin supplement, 14.

### 2.1. Milk Collection and Production of Dairy Derivatives

Cows were mechanically milked twice daily, at 6:00 a.m. and 5:00 p.m. Three separate milk collections were performed (for cheese and cream productions and for milk analysis). Milk collection involved milk from two consecutive days and from the 20 cows in each group to minimize daily and individual variability. Milk collected on both days was used separately, resulting in duplicate samples.

The first collections occurred on days 15 and 16, when 80 L/day/group were collected separately from the bulk tank for colonial cheese production. On each day, the milk was transported to Casa Bianchi Indústria e Comércio de Alimentos Ltda. (Lajeado Grande, SC, Brazil), a company specialized in dairy processing. Milk underwent low-temperature, long-time pasteurization (LTLT) and was subsequently used for colonial cheese production, with 80 L per group yielding approximately 12 kg of colonial cheese (~1000 g/peace, not individually determined), each day. Colonial cheeses were matured up to 60 days in controlled chambers (SuckMilk Ltda., Nova Erechim, SC, Brazil) at a mean moisture of 54.8 ± 16.89% and a temperature of 8.7 ± 2.84 °C, following Decree No. 362 [15].

During maturation, cheese samples in each group (control and EOB) (~250 g from each day) were collected on days 7, 20, and 45 for the determination of moisture [14], protein [16], ash [17], pH [17], fat [18], and thiobarbituric acid reactive substances (TBARS) [19]. Additionally, samples collected at 7, 20, 45, and 60 days of maturation were analyzed for fatty acid profiles [20] and microbiome composition.

The second milk collection took place on days 20 and 21, with 100 L of milk (50 L/day/group) transported to the Food Laboratory of the Serviço Nacional de Aprendizagem Industrial (SENAI), Chapecó, SC, Brazil. Milk was subjected to LTLT pasteurization and processed for cream extraction using an electric cream separator (Casa das Desnatadeiras Ltd., Goiânia, GO, Brazil; 50 L/h capacity). Cream samples were analyzed for acidity [21], fat [22], protein [16], ash [21], and TBARS [19].

Cream was stored in sterile, sealed containers (4 kg capacity) at ≤5 °C [23] until sensory evaluation. Cheese samples (days 7, 20, and 45) and cream samples were microbiologically evaluated for mold and yeast counts [21], coagulase-positive *Staphylococci* count [24], detection of *Salmonella* spp. [21], total *Escherichia coli* count [21], and detection of staphylococcal enterotoxins [21]. These microbiological analyses were performed to follow Ordinance No. 146 [23] and Regulatory Instruction No. 161 [25].

The third milk collection was conducted on days 30 and 31, with 20 L of milk (10 L/day/group) collected and transported to the SENAI Food Laboratory, Chapecó, SC, Brazil. Milk was pasteurized (LTLT) and analyzed in duplicate for density [26], defatted dry extract [26], total solids extract [26], fat [22], cryoscopy index [27], protein [16], acidity [26], and TBARS [19]. To ensure microbiological safety for sensory evaluations, milk samples were assessed for total enterobacteria counts, according to Regulatory Instruction No. 161 [25].

### 2.2. Sensory Evaluation

Sensory evaluations of milk, cream, and cheese were conducted at the Animal Science Department of Santa Catarina State University, Chapecó, SC, and at the Sensory Analysis Laboratory of SENAI, Chapecó. Milk and cream were evaluated once, while cheese was evaluated at 20 and 45 days of ripening. A total of 120 adult participants (≥18 years old), including both male and female dairy consumers, took part in the evaluations after signing an informed consent form. The analyses were conducted in a clean, well-lit, ventilated environment, free from distractions, using uniform containers and sample sizes, in accordance with current standards [28]. These procedures were in accordance with the Human Research Ethics Committee (CEPSH/UDESC-CAAE: 67158823.8.0000.0118).

In the first sensorial analysis, panelists received six coded samples: two of pasteurized milk (control and EOB; 30 mL each, stored at 4 °C for 48 h post-collection) and two of cream (control and EOB; 30 mL each, stored at 5 °C for 5 days post-production), served in plastic cups. Additionally, they received two ~25 g portions of colonial cheese (control and EOB) matured for 20 days, served on coded plastic plates. On the second sensorial analysis day, the same number of panelists evaluated colonial cheese samples (control and EOB) matured for 45 days, as described before.

For each sample, panelists were instructed to observe, smell, and taste before completing three tasks: (1) attribute evaluation based on Meilgaard et al. [29]; (2) purchase intention assessment according to Dutcosky [30]; and (3) product characterization using the check-all-that-apply (CATA) method. The sensory attributes were selected based on published studies involving similar products [31]. After the sensory evaluations, participants completed a questionnaire regarding their milk, cream, and cheese consumption habits and frequencies.

### 2.3. Microbiome Analysis of Colonial Cheese: Sample Processing, DNA Extraction, Library Preparation, and 16S rRNA Sequencing

To characterize the microbiome of colonial cheese during the maturation process, samples were collected on days 7, 20, 45, and 60. Wedge-shaped aliquots of 25 g were collected in duplicate (totaling 50 g) from each group (control and EOB). The samples were then transferred to sterile plastic containers, properly labeled, and stored at –18 °C for up to 30 days until DNA extraction. The DNA extraction protocol used was previously described by Endres et al. [32].

The samples were thawed at refrigeration temperature (approximately 4 °C) and individually ground using a food processor. From each homogenized sample, a 25 g subsample was taken and diluted in 225 mL of sterile distilled water. This mixture was homogenized in a shaker incubator at 110 rpm for 2 h without temperature control. The resulting suspension was filtered through sterile gauze to separate the sediment. A total of 35 mL of the filtrate was then centrifuged at 10,000 rpm for 40 min at 4 °C. The supernatant (~30 mL) was discarded, and the resulting pellet was resuspended in 3 mL of sterile distilled water and vortexed. Subsequently, 1 mL of this suspension was transferred to a microcentrifuge tube and centrifuged at 14,000 rpm for 5 min at 4 °C to form a new pellet.

This pellet was used for DNA extraction with the PureLink Genomic DNA Isolation Kit (Invitrogen, Carlsbad, CA, USA). The extracted DNA was eluted in 25 μL of ultrapure deionized water and stored at −20 °C until further analysis.

Microbial libraries were generated by amplifying the V4 region of the bacterial 16S rRNA gene using primers F515 and R806, both modified with Illumina adapter overhangs. PCR amplification was carried out in 25 μL reaction volumes containing 12.5 ng of genomic DNA, 1.5 mM MgCl_2_, 0.2 μM of each primer, 200 μM of each dNTP, 2 U of Platinum Taq DNA Polymerase (Invitrogen, Carlsbad, CA, USA), and 1× reaction buffer. Amplification was performed in a BioRad MyCycler thermal cycler (BioRad, Hercules, CA, USA) under the following conditions: initial denaturation at 94 °C for 3 min, followed by 30 cycles of 94 °C for 30 s, 55 °C for 30 s, and 72 °C for 30 s, with a final extension at 72 °C for 5 min.

Amplicons were purified using Agencourt AMPure XP magnetic beads (Beckman Coulter, Indianapolis, IN, USA), and dual-index barcodes were added to the purified products following the manufacturer’s protocol (Illumina Inc., San Diego, CA, USA). Sequencing was performed as described by Endres et al. [32].

Initial quality control of raw sequencing reads included the removal of low-quality sequences and chimeras. Subsequent bioinformatic analysis involved clustering the sequences into operational taxonomic units (OTUs) and generating taxonomic distribution profiles to characterize the microbial community composition.

### 2.4. Microbiome Analysis of Colonial Cheese: Bioinformatics Analysis

The sequencing data were processed using the QIAGEN CLC Genomics Workbench software pipeline version 23.0.4, according to the manufacturer’s instructions. A quality filter with a Phred score cut-off < 20 was applied to remove low-quality reads, in addition to the identification and removal of chimeric sequences. After trimming and cutting, taxon clustering was performed, generating taxonomic distribution graphs based on operational taxonomic unit (OTUs), using a 99% identity level. Sequences were classified based on the SILVA v138 database. With the generated data, the microbial communities, alpha diversity, beta diversity, and differential abundance of the samples were observed.

### 2.5. Statistical Analysis

Forty dairy cows were randomly assigned in equal numbers to two groups (control and EOB) using a completely randomized design. The physicochemical properties of milk, cream, and colonial cheese were presented in a descriptive table, as the limited number of replicates (n = 2) did not allow for robust statistical analysis. Fatty acid and TBARS data were analyzed by ANOVA using the SAS^®^ statistical package (version 9.2), following verification of the residual normality and homogeneity of variances. Once these assumptions were met, a repeated measures analysis of variance over time was performed. Means were compared using the Fisher–Snedecor test and were considered significantly different at *p* < 0.05.

Sensory analysis data were evaluated by ANOVA using XLSTAT software (Windows version 2012.5, Addinsoft, Paris, France). Means were compared using Tukey’s test, with significance set at *p* < 0.05.

The frequency of use for each CATA term was determined by counting the number of participants who selected that term to describe each sample (milk, cream, and cheese at 20 and 45 days of maturation) in both groups (control and EOB). Cochran’s *Q* test was applied to identify significant differences in the term frequencies used to describe the attributes of milk and its derivatives. Subsequently, Marascuilo’s procedure [33] was used for a pairwise comparison of the cited attributes for each sample and group. In both tests, means were considered significantly different when *p* < 0.05.

The microbial alpha diversity of the samples was determined using abundance-based coverage estimation (ACE), Chao1 diversity index and Shannon index. For statistical comparisons of these indexes, a Mann Whitney test was applied. Beta diversity was evaluated through principal coordinates analysis (PCoA) based on Bray–Curtis dissimilarity and weighted UniFrac distances, which incorporate both taxonomic and phylogenetic differences among microbial communities. PCoA ordination was applied to visualize clustering patterns in beta diversity. To statistically assess the significance of the group separations observed in the PCoA plots, a permutational multivariate analysis of variance was conducted on the corresponding distance matrices.

## 3. Results

In the appendix section, Appendix A summarizes the descriptive values of the physicochemical and microbiological parameters of milk, cream, and colonial cheeses produced from cows with or without EOB addition in the diet. Although the absence of statistical analysis limits the strength of the conclusions, the data suggests that EOB supplementation did not markedly influence the physicochemical or microbiological characteristics of milk and cream from Jersey cows. Regarding colonial cheeses (Appendix A), a progressive reduction in moisture content was observed during the ripening period, decreasing from 47.34% to 35.61% in the Control group and from 46.74% to 34.50% in the EOB group, based on measurements taken at 7 and 45 days of maturation. No changes were detected in the microbiological profile of the cheeses throughout ripening or in response to EOB supplementation.

Thiobarbituric acid reactive substance (TBARS) values of the colonial cheese were significantly influenced by both the treatment with the essential oil blend (EOB) and the interaction between treatment and ripening time (*p* = 0.0106 and *p* = 0.0003, respectively) (Table 1). At 7 and 20 days of ripening, no significant differences were observed between the control and EOB groups. However, at 45 days, cheeses from the control group showed higher TBARS values (0.1585) compared to those from cows supplemented with EOB (0.1300), indicating a lower degree of lipid oxidation in the latter. The overall mean TBARS value was slightly lower in cheeses from the EOB group (0.1385) than in the control group (0.1448), suggesting a potential antioxidant effect of the essential oil supplementation.

Table 2 presents the mean values of the fatty acid profile of colonial cheese at four ripening stages (7, 20, 45, and 60 days). Supplementation with the essential oil blend (EOB) significantly influenced the fatty acid profile of colonial cheese produced from the milk of Jersey cows. Cheeses from cows supplemented with EOB showed higher levels of saturated fatty acids (SFA) compared to the control group (74.49% vs. 73.92%; *p* = 0.030). Conversely, the content of unsaturated fatty acids (UFA) was lower in the EOB group (25.50%) than in the control (26.07%; *p* = 0.030), with a notable reduction in monounsaturated fatty acids (MUFA), which were significantly lower in the EOB group (22.41% vs. 22.97%; *p* = 0.038).

Polyunsaturated fatty acid (PUFA) levels did not differ between treatments (*p* = 0.769). However, a significant interaction between treatment and cheese maturation time was observed (*p* = 0.015), with a reduction in PUFA content at 60 days in the EOB group. The UFA/SFA ratio was also lower in the supplemented group (0.34 vs. 0.35; *p* = 0.029), indicating a higher proportion of saturated fat in these cheeses.

No significant differences were found between treatments for total omega-6 (*p* = 0.925) or omega-3 (*p* = 0.284) fatty acid contents. However, a significant treatment × maturation time interaction was observed for omega-3 fatty acids (*p* = 0.021), with a lower value at 60 days in the EOB group. The ω6/ω3 ratio was higher in the EOB group at 60 days (*p* = 0.016), although the overall difference between treatments was not significant (*p* = 0.099).

### 3.1. Sensory Evaluation of Milk, Cream, and Colonial Cheese

Regarding the sensory analysis, the panel consisted of 37% male and 63% female participants, with a mean age of 29 ± 8 years. Participants were also characterized based on their milk consumption habits, with the majority reporting regular intake of whole milk (78%) and consumption more than once a week (65%), indicating that milk is a high-frequency dietary item for this population. The majority of sensory test participants reported consuming full-fat cream (84%), with varying consumption frequencies: more than once a week (11%), once a week (39%), once a month (38%), and rarely (12%).

Hedonic evaluation revealed no significant differences in milk texture (7.1 ± 1.6 vs. 7.4 ± 1.6) or acidity (6.6 ± 2.2 vs. 7.0 ± 2.1) between the control and EOB group samples. However, the treated sample received significantly higher scores for flavor (6.6 ± 1.8 vs. 7.4 ± 1.5) and overall impression (6.9 ± 1.7 vs. 7.4 ± 1.7) (*p* < 0.05). Despite these differences in sensory acceptance, purchase intention scores for the control and EOB samples showed only a marginally significant difference (*p* = 0.081), with respective responses as follows: texture (24 vs. 39), flavor (37 vs. 35), acidity (22 vs. 17), and overall impression (17 vs. 9).

A hedonic evaluation of the cream samples revealed that, in terms of texture, samples from the EOB group were significantly preferred (7.4 ± 1.4) compared to those from the control group (6.7 ± 1.9) (*p* < 0.05). For the other hedonic parameters, no significant differences were observed (*p* > 0.05). The flavor scores were 7.4 ± 1.5 and 7.1 ± 1.7. The acidity scores were 6.9 ± 2.1 and 6.7 ± 2.1, and the overall impression scores were 7.4 ± 1.4 and 7.1 ± 1.6 for the EOB and control groups, respectively.

Similarly, no significant differences were found in purchase intention (*p* > 0.05), with 35% of participants indicating they would “definitely buy” the product and fewer than 14% responding that they would “probably buy or not buy” it, suggesting an overall favorable level of product acceptance.

The check-all-that-apply (CATA) results for the milk samples are presented in Table 3. Both the control and EOB samples were described similarly across most sensory attributes, with the most frequently cited terms being “supermarket milk taste,” “raw milk taste,” and “sweet taste”—each cited by more than 23% of participants for both samples.

Despite this general similarity, the milk from animals supplemented with EOB was significantly more often described as “sweet” compared to the control (40% vs. 25% of responses, *p* = 0.004). Additionally, the term “watery” was cited less frequently for the treated sample (*p* = 0.040), while the term “opaque” showed a marginally significant difference, being more associated with the EOB milk (*p* = 0.071).

The CATA test results (Table 3) showed that the cream was described similarly across most attributes (*p* > 0.05). However, the cream from the EOB group was more frequently perceived as similar to commercial cream (*p* = 0.005).

The sensory evaluation of colonial cheese was conducted after 20 and 45 days of maturation. At 20 days, 37% of the participants were male and 63% female, with a mean age of 29 ± 8 years. At 45 days, the sample consisted of 64% female and 36% male participants, with a mean age of 34 ± 9 years. Participants’ preferences and cheese consumption habits revealed that the majority preferred semi-fat cheese (74%), with preferred textures ranging from semi-soft (42%) to firm (31%) and soft (27%). Most participants favored cheeses with a slightly strong flavor (61%) and of the colonial type (67%). Additionally, 70% reported consuming cheese more than once a week, indicating that cheese is a regular part of their diet.

No significant differences were observed in the hedonic evaluation of colonial cheese in terms of texture, flavor, acidity, or overall impression between the control and EOB groups at the same maturation time, nor between the samples evaluated at different maturation times. Similarly, purchase intention did not differ significantly between samples after 20 and 45 days of maturation (*p* > 0.05), with approximately 57% of participants indicating that they would definitely or probably purchase any of the cheeses.

The CATA test responses for cheese samples produced from the milk of the control and EOB groups, evaluated after 20 and 45 days of maturation, indicated that the cheeses were perceived as similar across most of the sensory attributes assessed (Table 4). Despite the overall similarity in the most frequently cited terms, the 20-day matured cheese from the EOB group was more frequently described as “yellow” compared to the control (13% vs. 7% of responses, *p* = 0.046).

For the samples matured for 45 days, the cheese produced from EOB milk was more often associated with the attributes “greasy” (15% vs. 8%, *p* = 0.021) and “spicy” (10% vs. 4%, *p* = 0.035), although the overall frequency of citation for these terms remained relatively low (< 15%). Attributes indicating off-flavors (e.g., grassy, off-flavor, barn/silage, and herbal) were cited by fewer than 5% of participants, with no significant differences among samples. Given that only a few attribute citations reached statistical significance, constructing a perceptual map (CATA figure) was considered unnecessary.

### 3.2. Microbial Composition of Colonial Cheese

The microbial composition of the cheese samples was analyzed through high-throughput sequencing of the hypervariable V4 region of the *16S rRNA* gene. A total of 4561 operational taxonomic units (OTUs) were identified across eight samples, comprising four cheese samples derived from milk of the control group and four from the EOB-addition group at different ripening stages. Rarefaction curve analysis indicated saturation in all samples, confirming that sequencing depth was sufficient to capture the microbial diversity present.

At a significance level of 5%, no statistically significant differences in relative microbial abundance were observed between the control and EOB groups. Bacillota was the predominant phylum in all samples. At the class level, Bacilli were most abundant, while the orders Lactobacillales and Staphylococcales were dominant. The most abundant families detected were Streptococcaceae in both treatment groups. The most frequently identified genera included *Lactococcus*, *Streptococcus*, and *Staphylococcus*. Among these, *Lactococcus* was the most prevalent genus in both the control and treated samples, followed by *Staphylococcus*. *Streptococcus* spp. were detected exclusively in cheese samples from the control group. Nevertheless, specific species such as *Staphylococcus equorum*, *Streptococcus salivarius*, and *Lactococcus lactis* were identified in both the control and EOB.

Overall, the microbial community composition was largely similar between groups, with Bacillota and the genera *Lactococcus*, *Streptococcus*, and *Staphylococcus* dominating the cheese microbiota, as illustrated in Figure 1.

Alpha diversity data were estimated using the Chao1 and Shannon indices. Regarding the diversity of colonial cheese, no differences were observed between the control and EOB cheese samples (*p* > 0.05). Alpha diversity indices, such as Chao1 and Shannon (Figure 2), are widely used to assess species richness and diversity in microbiological samples. In the context of cheese production, microbial diversity is a crucial factor influencing the safety, sensory quality, and functional properties of the product. The absence of significant differences in alpha diversity between colonial cheese samples with and without EOB suggests that the addition of EOB did not significantly alter the richness and evenness of the microbial communities present in the cheese.

Beta diversity evaluates differences in operational taxonomic unit (OTU) composition between samples. In this study, principal coordinates analysis (PCoA) was performed based on the relative abundance of OTUs to compare microbial community structures between control and essential oil EOB milk samples, using Bray–Curtis dissimilarity and weighted UniFrac distance metrics. No statistically significant differences in microbial composition were observed between the control and EOB groups, with *p*-values of 0.22 for Bray–Curtis and 0.11 for weighted UniFrac (Figure 3). Additionally, the maturation period did not significantly influence microbial community composition across the evaluated samples. The dots in the figures represent the individual values of the alpha diversity samples (species richness and evenness) in differ-ent groups (CONTROL vs. EOB). The similar distribution of points between the groups and the *p*-value of 0.08 (not significant, *p* > 0.05) indicate that there is no statistically relevant difference in microbial diversity between the conditions analyzed. This sug-gests that the EOB treatment did not significantly alter alpha diversity compared to the control.

Regarding differential abundance, a lower overall abundance of microbial taxa was observed in cheese samples produced from EOB milk, as illustrated in Figure 4 and Figure 5.

## 4. Discussion

At the outset, it is important to clarify that the objective of this study was not to evaluate the progressive effects of EOB supplementation over time, but rather to assess the impact of supplementation at specific time points. The experimental design considered targeted supplementation periods prior to each dairy product collection: cows were supplemented with EOB for 15 days before milk collection for colonial cheese production, 20 days before cream production, and 30 days before milk analysis. This approach allowed us to investigate whether the inclusion of EOB in the diet could influence the microbiological and physicochemical characteristics of milk and its derivatives after defined supplementation periods. We acknowledge that the modulation of the intestinal microbiota by EOB may require a longer adaptation period, and therefore, we recommend that future studies explore the time-dependent effects of EOB supplementation on gut microbiota and its potential impact on milk quality.

This EOB has previously been tested on calves [12] and piglets [13]. However, for its application in dairy cows, it is initially necessary to verify whether its use alters the characteristics of milk and its by-products. From a comprehensive perspective, this study encompassed all stages of dairy production. We began by supplementing the diet of Jersey cows with an EOB to assess whether this supplementation would influence the physicochemical and sensory characteristics of dairy products (milk, cream, and colonial cheese). More specifically, we evaluated the fatty acid profile and lipid oxidation of cheese, alongside a deep exploration of its microbiome. This research was motivated by the distinct properties of the essential oils used, notably their bactericidal effects [9,10], antioxidant activity [7,9,10], and strong aromatic potential [8]. EOB supplementation modulated the fatty acid profile and reduced lipid oxidation in colonial cheese without negatively affecting the sensory attributes of the cream or cheese.

The physicochemical analyses of colonial cheese complied with current legislation, both at the national level [26] and specific state regulations in Santa Catarina, including Decree 2.197/2022 [34] and Decree 362/2019 [14], which regulates Law 17.486/2018 for colonial cheese. The evaluated parameters included fat, protein, total solids extract, defatted dry extract, density, acidity, and cryoscopic index for milk, as well as fat, protein, ash, and acidity for cream. Microbiological evaluations included total enterobacteria count in milk and coagulase-positive staphylococci, mold and yeast count, Salmonella spp. detection, Escherichia coli count, and staphylococcal enterotoxin detection in cream. All milk, cream, and cheese samples, from both treatments (with or without EOB supplementation), met legal specifications according to the permissible limits listed in Appendix A.

These analyses were conducted in certified laboratories following international quality standards, in accordance with Brazilian legislation. The number of samples submitted followed legal requirements, which, although insufficient for statistical comparison, allowed descriptive presentation (Appendix A). Since all products met microbiological standards, sensory evaluations were conducted on milk, cream, and colonial cheese at 20 and 45 days of ripening.

Campanha et al. [35] similarly evaluated the addition of essential oils (carvacrol, cinnamaldehyde, eugenol, and capsicum oleoresin) to the diet of crossbred cows and found no significant differences in the physicochemical milk parameters between the control and treated groups (5 g/cow/day of EOB). These results suggest that EOB supplementation may not significantly affect the macroscopic composition of milk.

There is currently no national regulation in Brazil defining the identity and quality of colonial cheese, resulting in considerable variability in its characteristics depending on the production region [36]. Nevertheless, physicochemical data, along with sensory and consumer acceptance characteristics, are important for product classification and market understanding [37]. According to Ordinance No. 146/1996 [23], colonial cheeses in this study—produced with or without EOB—ranged from high to medium and, later, low moisture content, with values ranging from 47.34% to 34.50% between 7 and 45 days of ripening. Moisture content can be influenced by ripening time and rind formation, as well as pressing pressure and salting method. Roldan et al. [38] reported an average moisture content of 46.5% in colonial cheese, classifying it as high-moisture cheese, typically due to short ripening periods. In the present study, cheese at 7 days of ripening fits this classification.

Fat content ranged from 28.00% to 35.25%, classifying the cheese as semi-fat, as per [23], given the moisture range of 25.0% to 44.9%. In contrast, ref. [37] reported lower values (19.33% to 26.95%) for colonial cheeses in Paraná. The higher values observed here may be attributed to the higher milk fat content of Jersey cows (3.7% to 3.9%; Appendix A). Similarly, the cheese protein content ranged from 18.91% to 25.07%, consistent with findings from [37], who reported 16.74% to 28.35%.

During ripening, pH values ranged from 5.49 to 6.08. Although there is no specific national regulation for colonial cheese identity and quality, it is known that pH should be below 6.0 at the end of ripening, especially for raw milk cheeses, to prevent growth of pathogens.

EOB supplementation significantly reduced lipid oxidation in colonial cheese throughout ripening. At 45 days of ripening, cheeses from the EOB group had lower TBARS values (0.1300 vs. 0.1585 mg MDA/kg; *p* = 0.0106). The perception threshold for rancid flavor ranges from 0.5 to 2.0 mg MDA/kg [39]. Although detectable, TBARS levels observed here were below this threshold and did not negatively affect sensory attributes.

TBARS values < 0.2 mg MDA/kg are considered low and indicate oxidative stability [40], enhancing food safety due to the toxic and mutagenic potential of malondialdehyde at high concentrations [41]. Lower TBARS values are linked to better preservation of desirable sensory traits, including texture, flavor, aroma, and color, as oxidation-derived volatiles are minimized.

These results align with other findings of this study. Despite the increase in saturated fatty acids (SFA) in cheeses from EOB-supplemented cows (Table 2), the TBARS values remained low. This lipid profile change did not compromise the sensory quality of cheese or other dairy products, as shown in sensory analyses (Table 3 and Table 4).

EOB supplementation significantly altered the fatty acid profile of colonial cheese, increasing SFA and reducing UFA, especially MUFA. These effects likely result from bioactive compounds in essential oils modulating rumen microbial activity and favoring fatty acid saturation [41].

While total PUFA levels did not differ significantly, there was a treatment–ripening time interaction, with a PUFA reduction at 60 days in the EOB group. This may reflect higher PUFA susceptibility to oxidation during maturation, which can be mitigated by antioxidant compounds in essential oils [10].

The UFA/SFA ratio was lower in EOB cheeses, indicating a higher saturated fat content. Although SFAs are often linked to cardiovascular risk, the specific profile is relevant—e.g., stearic acid (C18:0) is neutral regarding blood cholesterol. Additionally, bioactive compounds in essential oils (such as phenols, flavonoids, and tocopherols) may provide functional benefits that offset the SFA increase.

No significant differences in omega-3 fatty acid levels were found. However, interaction effects showed lower omega-3 at 60 days in EOB cheeses. Consequently, the ω6/ω3 ratio was higher in this group. A balanced omega-6/omega-3 ratio is desirable in the human diet due to anti-inflammatory and chronic disease prevention effects [42].

Essential oil supplementation positively influenced milk and cream sensory attributes, particularly flavor and texture, and maintained cheese acceptability despite changes in fatty acid profile and reduced lipid oxidation. These results suggest that natural compounds like menthol and eucalyptol may enhance sensory stability, likely due to their antioxidant properties [7,10]. The absence of undesirable flavors and preservation of expected sensory characteristics support the technological and commercial potential of this supplementation strategy.

### 4.1. Microbiome

Among these, *Lactococcus* was the most prevalent genus in both the control and treated samples, followed by *Staphylococcus*. Microorganisms of the genus *Staphylococcus* are known for their ability to produce heat-stable enterotoxins, which can cause foodborne intoxication in consumers, representing a significant risk to food safety. *Streptococcus* spp. were detected exclusively in cheese samples from the control group. Species-level resolution was limited due to the partial sequencing of the *16S rRNA* gene, which lacks the discriminatory power to identify many bacterial species [43]. Nevertheless, specific species such as *Staphylococcus equorum*, *Streptococcus salivarius*, and *Lactococcus lactis* were identified in both control and EOB-treated samples.

The predominant presence of *Bacillota* in cheese samples is corroborated by several studies investigating the microbial composition of different types of cheese. *Bacillota* is a phylum widely recognized for its significant role in the fermentation of dairy products. This phylum includes many genera of lactic acid bacteria, such as *Lactococcus*, *Lactobacillus*, *Streptococcus*, and *Enterococcus*, which are essential for cheese production and maturation. Quigley et al. [44] studied the microbiota of different Irish artisanal cheeses and observed that *Bacillota* was the predominant phylum, with *Lactobacillus* and *Streptococcus* being the most common genera. This study highlighted the importance of these bacteria in defining the organoleptic characteristics of cheeses, such as texture and flavor, through the production of lactic acid and other metabolic compounds.

Another study conducted by Wolfe et al. [45] analyzed the microbial ecology of maturing cheeses and found that *Bacillota* constituted the majority of the bacterial community. They observed that, in addition to lactic acid bacteria, other members of *Bacillota*, such as *Staphylococcus* and *Bacillus*, also played a crucial role in cheese maturation, influencing crust formation and internal maturation.

Research on artisanal cheeses from the southeastern region of Brazil identified microorganisms belonging to four microbial phyla: *Acinetobacter*, *Bacteroidetes*, *Bacillota*, and *Proteobacteria*, and 109 genera. Of this amount, 31 genera stood out, including 9 genera considered natural to dairy samples and 22 pathogenic or harmful to dairy technology [46]. Researchers investigating cheeses in the southwestern region of Brazil found similar results; in general, *Lactococcus lactis* was the most abundant species, regardless of the cheese sample characteristics [47].

The genus *Streptococcus* was observed only in the cheese samples produced with control milk, and this absence of the genus *Streptococcus* in cheeses produced with milk treated with essential oils is a significant finding, as it suggests that these compounds can influence the microbial composition of the cheese. Several studies have investigated the impact of essential oils on the microbiota of dairy products, demonstrating that these natural compounds have antimicrobial properties that can alter the microbial community of milk and, consequently, cheese.

EOBs are known for their antimicrobial, antifungal, and antioxidant properties. They contain bioactive compounds such as terpenes, phenols, and aldehydes, which are effective against a wide range of microorganisms. In cheese production, the addition of EOB to milk can inhibit the growth of undesirable bacteria and shape the maturation microbiota.

De Souza et al. [48] investigated the use of oregano and rosemary EOs in milk for the production of Minas Frescal cheese. They observed a notable reduction in the population of *Streptococcus* spp. and other pathogenic bacteria. Treatment with essential oils also promoted a favorable environment for beneficial bacteria, such as *Lactobacillus*, which are essential for fermentation and flavor development in cheese.

Pesavento et al. [49] explored the use of rosemary essential oil in goat milk cheeses and observed a significant decrease in *Streptococcus spp*. counts, while other genera, such as *Lactococcus* and *Enterococcus*, remained relatively unchanged. This study suggested that essential oils can be used to target the microbial composition without compromising the quality of the final product. EOs, due to their antimicrobial properties, have the potential to modify the microbiota of milk and cheese, reducing the presence of undesirable bacterial genera such as *Streptococcus* and favoring beneficial microorganisms that contribute to the sensory quality and safety of the final product. These findings are important for the dairy industry as they offer natural alternatives for microbial control during cheese production.

#### 4.1.1. Alpha Diversity

Alpha diversity indices, such as Chao1 and Shannon (Figure 2), are widely used to assess species richness and diversity in microbiological samples. In the context of cheese production, microbial diversity is a crucial factor influencing the safety, sensory quality, and functional properties of the product. The absence of significant differences in alpha diversity between colonial cheese samples with and without EO suggests that the addition of EO did not significantly alter the richness and evenness of the microbial communities present in the cheese.

Parvez et al. [50] investigated the microbial diversity of artisanal cheeses using high-resolution sequencing and alpha diversity analyses, including the Shannon and Chao1 indices. They observed that factors such as milk type and maturation conditions influenced microbial diversity, but specific interventions, such as the addition of natural preservatives, did not result in significant differences in alpha diversity. This finding is consistent with the results observed in colonial cheeses treated with EO, suggesting that these compounds do not drastically affect alpha diversity. Cocolin et al. [51] analyzed the microbiota of cheeses produced with different natural antimicrobial treatments, including essential oils. They also used alpha diversity indices and found that, although the specific microbial composition varied, the alpha diversity did not show significant differences between treatments. This supports the idea that the addition of EO can alter microbial composition without impacting overall diversity.

De Filippis et al. [52] reported in their study that natural interventions, such as the addition of EO, can influence the relative abundance of certain bacterial genera but do not necessarily affect alpha diversity. This result aligns with observations of colonial cheeses with and without EO. The available data indicate that the addition of EO to milk used for colonial cheese production does not significantly alter the alpha diversity of the present microbiota, as evidenced by the Chao1 and Shannon indices. Recent studies corroborate that, although essential oils can modify microbial composition, alpha diversity remains relatively stable, ensuring that the sensory characteristics and safety of the cheeses are not compromised.

#### 4.1.2. Beta Diversity

No statistically significant differences in microbial composition were observed between the control and EOB groups, with *p*-values of 0.22 for Bray–Curtis and 0.11 for weighted UniFrac (Figure 3). Additionally, the maturation period did not significantly influence the microbial community composition across the evaluated samples.

Endres et al. [32] investigated the beta diversity of cheeses produced with pasteurized milk and the addition of starter cultures, observing that these practices significantly influence microbial composition. The study used beta diversity analyses, including Bray–Curtis and UniFrac, and found clear differences between cheeses produced with different treatments. However, it is important to note that these differences may be more evident in highly controlled systems or with specific interventions, such as pasteurization and starter cultures, which shape the microbiota of the final product. Duarte et al. [53] analyzed the beta diversity of artisanal and industrial cheeses. They found that pasteurization and the addition of starter cultures significantly altered microbial composition, with artisanal cheeses showing greater microbiological variability compared to industrial ones.

De Filippis et al. [52] conducted a study on cheese maturation and its impact on beta diversity. They observed that maturation time and environmental conditions influence microbial composition, but interventions such as the addition of natural antimicrobials did not result in significant differences in beta diversity. These findings are in line with the present result, indicating that the addition of EO may not significantly impact microbial composition over the maturation period.

#### 4.1.3. Differential Abundance

Regarding the differential abundance, a lower overall abundance of microbial taxa was observed in cheese samples produced from EO-treated milk, as illustrated in Figure 4 and Figure 5. Differential abundance analysis is a critical approach to understanding how specific treatments influence the microbial composition of food products such as cheese. Different táxons were observed in cheese samples produced with milk treated with EOs; this can be attributed to the antimicrobial action of EOs, which are known for their inhibitory properties against a wide range of microorganisms.

Marcial et al. [54] conducted a study to evaluate the impact of different natural antimicrobial compounds, including oregano EO, on cheese microbiota. The authors observed that in the EO-treated samples, there was a suppression of spoilage microorganisms such as molds and yeasts, without compromising the viability of lactic acid bacteria starter cultures. Moreover, the addition of the oil did not negatively affect the sensory quality of the product. These findings are consistent with the results of the present study, suggesting a potential selective action of EO on cheese microbiota. Cui et al. [55] found in their study that plant EO, such as thyme and oregano oil, significantly reduced the abundance of undesirable bacteria, including *Staphylococcus* and *Escherichia coli* species, while maintaining or increasing the presence of beneficial bacteria such as *Lactobacillus*. This study supports the hypothesis that EOs have a specific antimicrobial effect, beneficially modulating the microbial composition.

Pinto et al. [56] highlighted that EOs can reduce the abundance of undesirable microorganisms in food, contributing to its preservation and safety. The bioactive compounds present in EOs primarily act by destabilizing the microbial cell membrane, leading to loss of structural integrity, leakage of intracellular contents, and consequently, inhibition of growth or cell death. However, in addition to inhibiting pathogens, EOs may also affect beneficial microorganisms, such as lactic acid bacteria, making it necessary to carefully adjust their concentrations to avoid negative impacts on the desired microbiota.

The application of EOB in milk used for cheese production can result in a significant differential abundance of microorganisms, generally reducing the presence of pathogenic and spoilage bacteria. The antimicrobial action of EO, which manifests through the disruption of cell membranes and inhibition of critical metabolic processes, explains the lower abundance observed in cheese samples produced with treated milk. These results reinforce the potential of EOs as natural antimicrobial agents in cheese production.

The study demonstrated that EO not only reduced the total bacterial count but also altered the microbiota composition, favoring the presence of specific lactic acid bacteria that contribute to the safety and sensory quality of the cheese.

## 5. Conclusions

The inclusion of eucalyptus and mint essential oils at the tested dosage did not induce perceptible alterations in the organoleptic properties of milk or its derivatives. Although the supplementation significantly modified the fatty acid profile, these changes did not impact consumer acceptability of the resulting dairy products, indicating that the characteristic aromas and flavors of the oils were either not transferred to the milk fat matrix or remained below sensory detection thresholds. Furthermore, the potential antimicrobial effects of the essential oils contributed to enhancing the microbiological safety of dairy products, as evidenced by the reduced presence of pathogenic microorganisms in the milk from supplemented cows.

## Figures and Tables

**Figure 1 foods-14-02788-f001:**
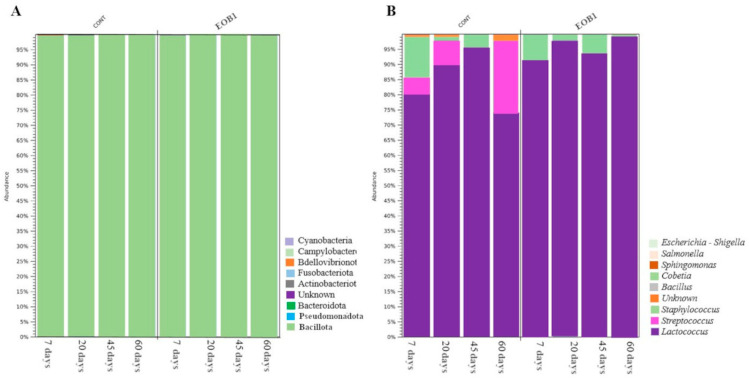
Relative abundance of bacterial taxa in colonial cheese samples, based on *16S rRNA* gene sequencing, at the phylum (**A**) and genus (**B**) levels. Samples were derived from cheeses produced with milk from Jersey cows fed with or without an essential oil blend (EOB). Vertical bars represent the mean relative abundance of bacterial sequences across replicates. Taxa that could not be classified are indicated in orange and labeled as “unknown”.

**Figure 2 foods-14-02788-f002:**
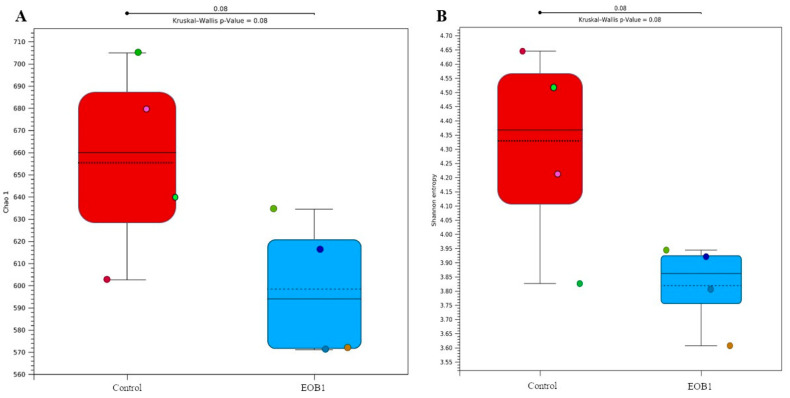
Alpha diversity metrics with control and treated milk samples. Richness measured by the Chao1 index (**A**) by the Shannon index (**B**). Samples were derived from milk obtained from Jersey cows fed with or without an essential oil blend (EOB). Dots represent individual alpha diversity values (species richness and evenness) for CONTROL and EOB groups. Similar distributions and a *p*-value of 0.08 (not significant, *p* > 0.05) indicate no statistically significant difference in microbial diversity between treatments, suggesting that EOB did not significantly affect alpha diversity compared to the control.

**Figure 3 foods-14-02788-f003:**
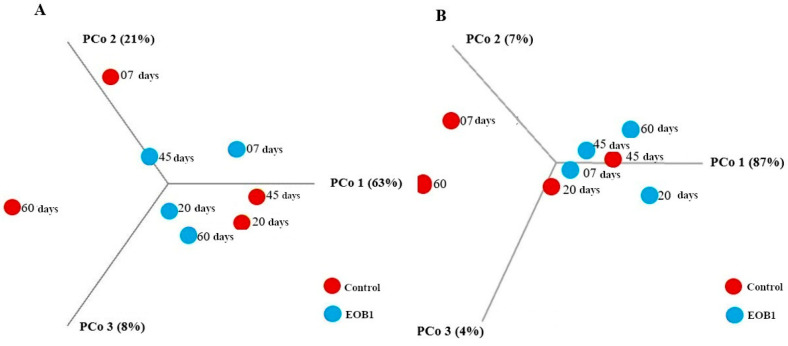
Beta diversity metrics of colonial cheese microbiota from Jersey cows fed with or without an essential oil blend (EOB): Bray–Curtis (**A**) and UniFrac (**B**) analyses. Red: Control; blue: EOB.

**Figure 4 foods-14-02788-f004:**
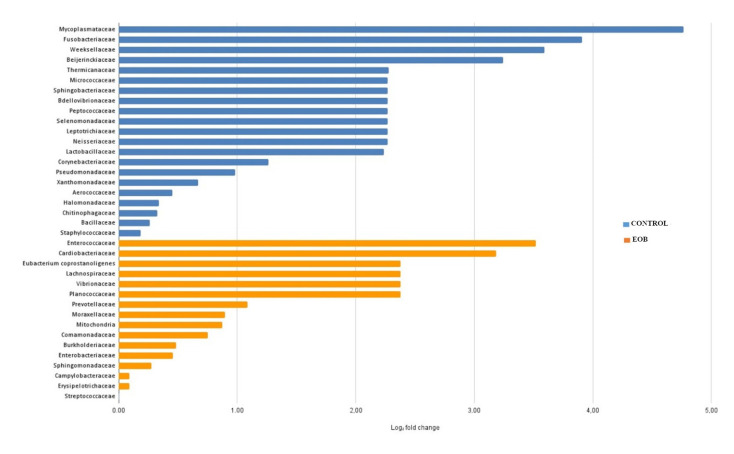
Differential abundance (Log_2_ fold change) of bacterial families in colonial cheese from Jersey cows fed with or without an essential oil blend (EOB^1^). Blue: Control; orange: EOB.

**Figure 5 foods-14-02788-f005:**
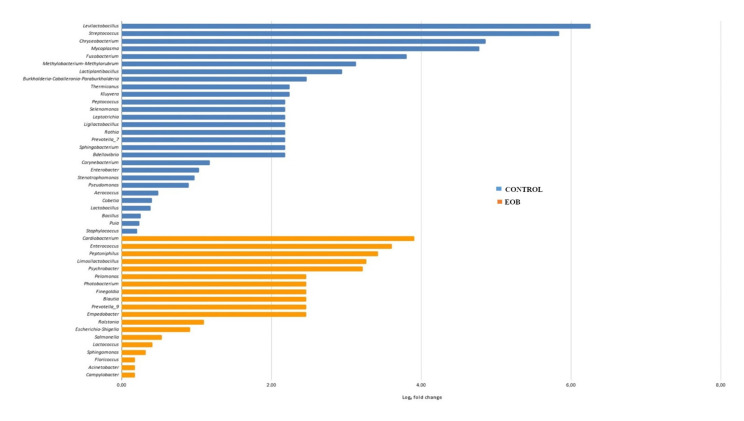
Log2 fold change: proportion of differential abundance at the genus level in colonial cheese from Jersey cows fed with (orange color) or without (blue color) an essential oil blend (EOB).

**Table 1 foods-14-02788-t001:** TBARS of colonial cheese obtained from the milk obtained from the milk of Jersey cows fed or not with essential oils blend (EOB ^1^) at 7, 20, and 45 days of maturation.

Days of Ripening	Treatments		*p*-Values	
	Control	EOB	T	T × D
7	0.1235	0.1230	0.0106	0.0003
20	0.1525	0.1625		
45	0.1585 a	0.1300 b		
Mean	0.1448	0.1385		

^1^ Essential oil blend composed of eucalyptus oil (157.9 g/L), peppermint oil (32 g/L), and menthol crystals (55 g/L) (BronchoVest, Biochem, Germany). Different letters within a column indicate significant differences (*p* < 0.05).

**Table 2 foods-14-02788-t002:** Fatty acid profile of colonial cheese obtained from the milk of Jersey cows fed (EOB ^1^) or not (Control) with essential oils blend at 7, 20, 45, and 60 days of maturation.

Fatty Acids, %	Treatments (T)	Cheese Maturation (CM), Days				Average ^2^	*p*-Value	
		7	20	45	60		*T*	*T* × *CM*
Saturated fatty acid (SFA)	Control	73.72	73.81	74.31	73.86	73.92 b	0.030	0.733
	EOB	74.36	74.64	74.47	74.49	74.49 a		
Unsaturated fatty acid (UFA)	Control	26.26	26.18	25.68	26.13	26.07 a	0.030	0.733
	EOB	25.63	25.35	25.52	25.50	25.50 b		
Monounsaturated fatty acid (MUFA)	Control	23.15	23.07	22.68	23.00	22.97 a	0.038	0.802
	EOB	22.50	22.21	22.44	22.50	22.41 b		
Polyunsaturated fatty acid (PUFA)	Control	3.12	3.10	3.00	3.13 a	3.09	0.769	0.015
	EOB	3.12	3.14	3.08	2.99 b	3.08		
UFA/SFA ratio	Control	0.35	0.35	0.34	0.35	0.35 a	0.029	0.738
	EOB	0.34	0.33	0.34	0.34	0.34 b		
Total ω6	Control	2.76	2.75	2.66	2.77	2.74	0.925	0.052
	EOB	2.77	2.78	2.73	2.66	2.73		
Total ω3	Control	0.30	0.30	0.29	0.32 a	0.30	0.284	0.021
	EOB	0.31	0.31	0.30	0.28 b	0.30		
ω6/ω3 ratio	Control	8.94	8.92	9.00	8.64 b	8.87	0.099	0.016
	EOB	8.86	8.86	9.00	9.33 a	9.01		

^1^ Essential oil blend composed of eucalyptus oil (157.9 g/L), peppermint oil (32 g/L), and menthol crystals (55 g/L) (BronchoVest, Biochem, Germany). ^2^ Different letters within a column indicate significant differences (*p* < 0.05)

**Table 3 foods-14-02788-t003:** CATA test of milk and cream obtained from Jersey cows fed or not with essential oils blend (EOB).

Attributes	Milk			Cream		
	Control	EOB ^2^	*p*-Value	Control	EOB	*p*-Value
Herbal taste	0.035	0.014	0.257	0.021	0.014	0.655
Sweet	0.246	0.401	0.004	0.261	0.232	0.505
Grass/countryside	0.120	0.077	0.201	0.028	0.042	0.527
Taste Weird	0.141	0.085	0.144	0.077	0.070	0.819
Tast Fatty	0.134	0.085	0.178	0.324	0.338	0.773
Bitter taste	0.035	0.021	0.480	0.021	0.014	0.655
Barn/silage	0.028	0.021	0.705	0.007	0.000	0.317
Rancid	0.042	0.021	0.317	0.035	0.007	0.102
Supermarket milk	0.268	0.324	0.317	n.d.	n.d.	n.d.
Watery	0.190	0.106	0.040	n.d.	n.d.	n.d.
Homogeneous	n.d. ^1^	n.d.	n.d.	0.190	0.225	0.369
Supermarket cream	n.d.	n.d.	n.d.	0.190	0.317	0.005

^1^ n.d. = not determined. ^2^ Essential oil blend composed of eucalyptus oil (157.9 g/L), peppermint oil (32 g/L), and menthol crystals (55 g/L) (BronchoVest, Biochem, Germany).

**Table 4 foods-14-02788-t004:** CATA test results for colonial cheese from Jersey cows fed with or without an essential oil blend (EOB ^1^) at 20 and 45 days of ripening.

Attributes	20 Maturation Days			45 Maturation Days		
	Control	EOB	*p*-Value	Control	EOB	*p*-Value
Fresh cheese	0.362	0.397	0.466	0.342	0.316	0.631
Fatty	0.064	0.099	0.225	0.079	0.149	0.021
Bitter taste	0.092	0.142	0.127	0.123	0.184	0.178
Barn/silage	0.021	0.014	0.655	0.009	0.000	0.317
Acid	0.234	0.241	0.873	0.228	0.211	0.732
Aged	0.106	0.121	0.683	0.237	0.254	0.655
Herbal flavor	0.007	0.007	1.000	0.018	0.053	0.157
Spicy	0.092	0.057	0.197	0.044	0.105	0.035
Common colonial cheese	0.404	0.383	0.680	0.421	0.368	0.330
Grass/countryside flavor	0.064	0.021	0.058	0.044	0.044	1.000
Yellow	0.071	0.128	0.046	0.035	0.053	0.414

^1^ Essential oil blend composed of eucalyptus oil (157.9 g/L), peppermint oil (32 g/L), and menthol crystals (55 g/L) (BronchoVest, Biochem, Germany).

## Data Availability

The original contributions presented in the study are included in the article/Appendix A, further inquiries can be directed to the corresponding authors.

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
