# Peer review of "Effects of Incorporation of Essential Oils in the Jersey Cow Diet on the Quality of Produced Dairy Products (Milk, Cream, and Colonial Cheese)"

_foods, 2025, doi:10.3390/foods14162788_

Round 1
Reviewer 1 Report
Comments and Suggestions for Authors
In this manuscript, jersey cows were fed with the given amounts of eucalyptus and peppermint essential oils and then their effects were determined on the physicochemical, microbiological, and sensory characteristics of obtained dairy products (milk, cream, and colonial cheese).
As it is obvious, this study is not just about sensory and microbial characteristics of cheese, so it can be suggested to change the title of the manuscript to cover all the subjects. It could be as below:
Effects of incorporation of essential oils in the Jersey cow diet on the quality of produced dairy products (milk, cream, and colonial cheese)
Abstract should be informative. Please include in line 25, the amount for EOB addition.
Line 28, Lower, How much?
In method section, line 98, 3.6 mL/cow/day, please include the reason for this amount, give reference or already preliminary study. It can be asked why not more or less!
All the results and discussion section are well, but there is one main question should be clarified and explained at the beginning of the discussion part. In this study cows were feed for some days with EO, so the milk and cream and cheese were obtained from which day of experiment and the presented results in Tables are related to which day? As you know it is matter of time that EO can affect the gut Microbia of the cows and the can affect the quality of milk and fat composition, so its effects are not so significant in the first days of experiment and will be pronounced after some times.
Table 1, title should be in English.
There is no need for 75 references in research paper, please give general and most related references particularly in introduction part and reduce it to 30 to 35.
Author Response
Please see the attachment.
For research article
Response to Reviewer X Comments |
||
1. Summary |
|
|
Thank you very much for taking the time to review this manuscript. Please find the detailed responses below and the corresponding revisions/corrections highlighted/in track changes in the re-submitted files. |
||
2. Questions for General Evaluation |
Reviewer’s Evaluation |
Response and Revisions |
We appreciate the time and effort dedicated to reviewing this manuscript. The Materials and Methods section has been enhanced, and additional information regarding the research design has been included. The presentation of the results has also been improved, particularly in the tables. 3. Point-by-point response to Comments and Suggestions for Authors |
||
Comments 1: As it is obvious, this study is not just about sensory and microbial characteristics of cheese, so it can be suggested to change the title of the manuscript to cover all the subjects. It could be as below: Effects of incorporation of essential oils in the Jersey cow diet on the quality of produced dairy products (milk, cream, and colonial cheese) Response 1: Thank you for your suggestion. We agree that the original title did not fully reflect the scope of the study. Therefore, we have revised the title of the manuscript as recommended to better encompass all evaluated parameters and dairy products. |
||
Comments 2: Abstract should be informative. Please include in line 25, the amount for EOB addition. |
||
Response 2: Thank you for your observation. We have revised the abstract to include the amount of EOB added, as suggested. The following information was added in line 25: “3.6 mL/cow/day”. Comments 3: Line 28, Lower, How much? Response 3: Thank you for your comment. We have revised the sentence to provide the specific values, as requested. The updated version now reads: “Additionally, EOB reduced lipid oxidation throughout the ripening process, with significantly lower TBARS values at 45 days of maturation (0.1300), compared to those from cows without supplementation of EOB (0.1585), suggesting improved oxidative stability.” |
||
Comments 4: In method section, line 98, 3.6 mL/cow/day, please include the reason for this amount, give reference or already preliminary study. It can be asked why not more or less! Response 4: Thank you for your observation. The dosage of 3.6 mL/cow/day was chosen based on the manufacturer’s recommendation. As noted in the revised manuscript, this information was added to clarify that the dosage of the EOB provided followed the instructions indicated on the commercial product label. |
Comments 5: All the results and discussion section are well, but there is one main question should be clarified and explained at the beginning of the discussion part. In this study cows were feed for some days with EO, so the milk and cream and cheese were obtained from which day of experiment and the presented results in Tables are related to which day? As you know it is matter of time that EO can affect the gut Microbia of the cows and the can affect the quality of milk and fat composition, so its effects are not so significant in the first days of experiment and will be pronounced after some times.
Response 5: Thank you for your thoughtful comment. We agree that the timing of sample collection in relation to the EOB supplementation period is an important point to clarify. In response, we have added the following explanation at the beginning of the Discussion section:
“At the outset, it is important to clarify that the objective of this study was not to evaluate the progressive effects of essential oil blend (EOB) supplementation over time, but rather to assess the impact of supplementation at specific time points. The experimental design considered targeted supplementation periods prior to each dairy product collection: cows were supplemented with EOB for 15 days before milk collection for colonial cheese production, 20 days before cream production, and 30 days before milk analysis. This approach allowed us to investigate whether the inclusion of EOB in the diet could influence the microbiological and physicochemical characteristics of milk and its derivatives after defined supplementation periods. We acknowledge that the modulation of the intestinal microbiota by EOB may require a longer adaptation period, and therefore, we recommend that future studies explore the time-dependent effects of EOB supplementation on gut microbiota and its potential impact on milk quality.”
We hope this addition addresses your concern and clarifies the study design.
Comments 6: Table 1, title should be in English.
Response 6: Thank you for your observation. We have translated the title of Table 1 into English, as requested.
Comments 7: There is no need for 75 references in research paper, please give general and most related references particularly in introduction part and reduce it to 30 to 35. Response 7: Thank you for your suggestion. We have reduced the number of references from 75 to 56, prioritizing the most relevant and general sources, especially in the Introduction section. Additionally, approximately 33% of the remaining references are cited in the Materials and Methods section to support the description of experimental procedures and analytical techniques. |
4. Response to Comments on the Quality of English Language |
Point 1: There was no. |
Response 1: There were no comments about the English language, but we would like to mention that the text has undergone an English revision. |
5. Additional clarifications |
We appreciate the comments. The adjustments have been made to the text, and the comments have been addressed. We hope to have clarified the doubts regarding the statistics. |

Reviewer 2 Report
Comments and Suggestions for Authors
Cows that graze freely have access to a diet of flowers and herbs when they are in bloom. Essential oils and other compounds found in these plants can naturally enter the bloodstream and contribute to changes in the composition of raw milk, thus influencing the production of dairy products.
Nevertheless, I find the manuscript interesting and useful because it provides information on how feeding practices (in this case, unilateral and controlled) can affect raw milk and consequently local traditional products. From a commercial perspective, the impact of essential oils on the production of fermented products and cheeses using propionic bacteria should be thoroughly analysed.
One strength of this manuscript is the detailed description of the methodology and results. The manuscript is coherent and accurate. I believe the topic 'Sensory evaluation of dairy products and microbiota characterisation of colonial cheeses produced from Jersey cow milk with or without the addition of essential oils to the diet' is very well written.
I would like the Authors to:
- In line 48: 'Recent studies have shown significant improvement' – please provide references.
- In line 190, 'room temperature' – what exactly is meant by this? Please specify the exact temperature.
- Results: appropriate sections should be added to separate the obtained results.
- A general note should also be included explaining why these particular EOBs were chosen and not others.
Author Response
Please see the attachment.
For research article
Response to Reviewer X Comments |
||
1. Summary |
|
|
Thank you very much for taking the time to review this manuscript. Please find the detailed responses below and the corresponding revisions/corrections highlighted/in track changes in the re-submitted files. |
||
2. Questions for General Evaluation |
Reviewer’s Evaluation |
Response and Revisions |
We appreciate the time and effort dedicated to reviewing this manuscript. The Materials and Methods section has been enhanced, and additional information regarding the research design has been included. The presentation of the results has also been improved, particularly in the tables. 3. Point-by-point response to Comments and Suggestions for Authors |
||
Comments 1: In line 48: 'Recent studies have shown significant improvement' – please provide references. Response 1: Thank you for your comment. We have added the appropriate references to support the statement in line 48, as requested. |
||
Comments 2: In line 190, 'room temperature' – what exactly is meant by this? Please specify the exact temperature. |
||
Response 2: Thank you for your observation. The sentence has been revised to clarify the temperature. The samples were thawed at refrigeration temperature (approximately 4 °C), and this information has been included in the manuscript. |
||
Comments 3: Results: appropriate sections should be added to separate the obtained results. Response 3: Thank you for your suggestion. To improve the organization and clarity of the Results section, we have added two subsections: 3.1. Sensory evaluation of milk, cream and colonial cheese 3.2. Microbial composition of colonial cheese We believe these additions help to better structure the presentation of the results. |
Comments 4: A general note should also be included explaining why these particular EOBs were chosen and not others.
Response 4: We thank the reviewer for the suggestion. We have added a general explanation regarding the choice of these particular EOBs at the beginning of the Discussion section.

Reviewer 3 Report
Comments and Suggestions for Authors
Thank you for your submission. I believe your research has certain value, but I still have some concerns:
1.The sample size for some indicators of milk, cream, etc. in the article is only 2 repetitions. Isn't that a bit small?
2. In the sensory evaluation, the "sweetness" of milk and the "commercial flavor" of cream in the EOB group received higher scores, but the specific definitions and quantitative standards of the evaluation indicators were not explained.
3. Distilled water was used in the control group. Although the volume was the same, it was not explained whether distilled water might have an impact on feed characteristics (such as palatability).
4. The study mentioned that EOB might reduce the abundance of pathogenic bacteria, but the specific quantity of common foodborne pathogenic bacteria in cheese was not detected.
5. Some studies cited in the discussion section (such as reference 63) have weak relevance to the microbial community of cheese.
6. The experimental period was only 31 days, and the long-term impact of EOB addition on cow health, milk production, and dairy product quality was not evaluated. It is recommended to explain the limitations of the study and propose the necessity of long-term follow-up experiments.
7. The article emphasizes the technical potential of EOB, but does not mention the addition cost, dose-effect relationship, and the feasibility of large-scale production.
Author Response
Please see the attachment.
For research article
Response to Reviewer X Comments |
||
1. Summary |
|
|
Thank you very much for taking the time to review this manuscript. Please find the detailed responses below and the corresponding revisions/corrections highlighted/in track changes in the re-submitted files. |
||
2. Questions for General Evaluation |
Reviewer’s Evaluation |
Response and Revisions |
We appreciate the time and effort dedicated to reviewing this manuscript. The Materials and Methods section has been enhanced, and additional information regarding the research design has been included. The presentation of the results has also been improved, particularly in the tables. 3. Point-by-point response to Comments and Suggestions for Authors |
||
Comments 1: The sample size for some indicators of milk, cream, etc. in the article is only 2 repetitions. Isn't that a bit small? Response 1: We appreciate the reviewer’s comment. As the main focus of this study at this stage was to conduct sensory analyses, only a limited number of samples were analyzed for the physicochemical parameters of milk, cream, and related products. These analyses served as a secondary objective, aimed at characterizing the samples prior to the sensory evaluation. For this reason, the detailed results of these physicochemical assessments were included as supplementary material rather than in the main body of the manuscript. |
||
Comments 2: In the sensory evaluation, the "sweetness" of milk and the "commercial flavor" of cream in the EOB group received higher scores, but the specific definitions and quantitative standards of the evaluation indicators were not explained. Response 2: Dear reviewer, thank you for your comment. The CATA attributes are nouns carefully chosen considering the expected description for the evaluated food and are commonly based on published studies involving similar products. In our study, the primary reference for selecting the terms was the manuscript published by Manzocchi et al (2021). Moreover, in the CATA method (Ares et al., 2010), participants receive only a list of attributes without definition and individually determine whether or not each term should be associated with the samples they tasted. Thus, there are no specific definitions or quantitative standards for CATA terms. - Manzocchi, E.; Martin, B.; Bord, C.; Verdier-Metz, I.; Bouchon, M.; De Marchi, M.; Constant, I.; Giller, K.; Kreuzer, M.; Berard, J.; Musci, M.; Coppa, M. Feeding cows with hay, silage, or fresh herbage on pasture or indoors affects sensory properties and chemical composition of milk and cheese. J Dairy Sci 2021 104. - Ares, G.; Deliza, R.; Barreiro, C.; Giménez, A.; Gámbaro, A. Application of a check-all-that-apply question to the development of chocolate milk desserts. J Sensory Studies 2010 25(1), 67–86. Comments 3: Distilled water was used in the control group. Although the volume was the same, it was not explained whether distilled water might have an impact on feed characteristics (such as palatability). Response 3: Thank you for your observation. We clarify that the use of distilled water in the control group does not interfere with the feed’s palatability. The small volume administered (3.6 mL/cow/day) was mixed with the TMR in both groups, and no differences in feed intake or animal behavior were observed throughout the experimental period. Comments 4: The study mentioned that EOB might reduce the abundance of pathogenic bacteria, but the specific quantity of common foodborne pathogenic bacteria in cheese was not detected. Response 4: Thank you for your contribution. I would like to clarify that, although our study observed a modulation in the microbial profile and performed some microbiological analyses of potentially pathogenic microorganisms in cheese, we did not detect a reduction in EOB-treated products. Therefore, although the results suggest the potential for EOB to influence the microbiota, we cannot confirm a reduction in pathogenic bacteria in this study. This point will be corrected to accurately reflect the data obtained. Comments 5: Some studies cited in the discussion section (such as reference 63) have weak relevance to the microbial community of cheese. Response 5: We appreciate the reviewer’s observation. We have reviewed the cited studies and removed or replaced those with limited relevance to the cheese microbial community, including reference 63. The discussion was adjusted to ensure better alignment with the objectives and scope of the study. Comments 6: The experimental period was only 31 days, and the long-term impact of EOB addition on cow health, milk production, and dairy product quality was not evaluated. It is recommended to explain the limitations of the study and propose the necessity of long-term follow-up experiments. Response 6: We thank the reviewer for this valuable observation. We acknowledge that the 31-day experimental period limits the evaluation of the long-term effects of EOB supplementation on cow health, milk production, and dairy product quality. However, this initial study primarily aimed to assess the feasibility of using EOB without altering the physicochemical characteristics of milk or being perceptible in sensory analyses. Complementary studies are currently underway to further investigate the long-term effects of EOB use, including its impact on animal health and performance. Additionally, it is worth noting that the use of EOB has also been proposed as a strategy to mitigate heat stress in dairy cows—an added potential benefit—which is typically managed through short-term interventions. Therefore, the short supplementation period applied in this study reflects both its practical use in the field and the initial focus on product integrity. Comments 7: The article emphasizes the technical potential of EOB, but does not mention the addition cost, dose-effect relationship, and the feasibility of large-scale production. Response 7: The present article aimed to evaluate the effects of EOB on dairy products (milk, cream, and Colonial cheese), including the fatty acid profile and the cheese microbiome, in addition to conducting a robust sensory analysis. Indeed, the technical potential of EOB as a feed additive for dairy cows—particularly its ability to mitigate heat stress and provide antioxidant benefits—was emphasized. However, we would like to clarify that the aspects mentioned by the reviewer—such as addition cost, dose-effect relationship, and the feasibility of large-scale production—will be comprehensively addressed in a separate article that is currently under preparation. This forthcoming manuscript will focus specifically on animal production, presenting data on performance, health, heat stress responses, and other related parameters. At present, these data are only available in the format of a master's thesis, which can be accessed at: https://sistemabu.udesc.br/pergamumweb/vinculos/0000b5/0000b5c8.pdf" It is important to note that the product evaluated in our study has previously been tested in pigs, poultry, and more recently, in dairy calves (Coelho et al., 2023), and it is already commercially available in several countries. Nonetheless, there are currently no published studies on its use in lactating dairy cows. The dosage applied in our experiment was based on a previously published study conducted with dairy calves, where the dose was calculated per kilogram of body weight and then extrapolated for adult cows. This dosage recommendation was established by the authors of the current study in consultation with technical staff from the partner company. We emphasize that none of the authors have any formal affiliation with the company that produces or sells the product. The company’s contribution was limited to technical advice, and this is duly acknowledged in the appropriate section of the manuscript. Therefore, while we fully acknowledge and agree with the reviewer’s observations, we respectfully submit that, considering the specific aims of the present study, the scope of the special issue (“Explore Milk and Dairy Products: Sensory, Physicochemical Characteristics, and Processing Technologies”), and the volume of data already included, it was more appropriate to address the animal production-related aspects in a separate manuscript. This parallel study is currently in development by our research group. Coelho, M.G.; da Silva, A.P.; de Toledo, A.F.; Cezar, A.M.; Tomaluski, C.R.; Barboza, R.D.F.; Virginio Júnior, G.F.; Manzano, R.P.; Bittar, C.M.M. Essential oil blend supplementation in the milk replacer of dairy calves: Performance and health. Plos One 2023, 18, e0291038. |

Reviewer 4 Report
Comments and Suggestions for Authors
This work studies the effects of the integration of essential oil (EO) blends in the rations of Jersey dairy cows in order to evaluate, the effects of the active ingredients present in the essential oils, on milk, cream and colonial cheese during maturation. The study involved physicochemical, microbiological and sensorial analyses on milk, cream and cheese. Furthermore, the cheese microbiome, the degree of lipid oxidation and the fatty acid profile were detected and evaluated. The results show that the essential oils integrated into the ration influenced the fatty acid profile with an increase in saturated fatty acids, the lipid oxidation was not significantly affected by the EO supplementation, the sensorial characteristics were not negatively affected and the study on the microbiome highlighted a potential antimicrobial effect of EO.
I would like to thank the authors for their interesting manuscript. The topic is interesting, and the work is generally well conducted and clearly presented.
Nevertheless, the manuscript needs some corrections and improvements, detailed below:
Most of the suggestions and corrections have been made directly in the draft PDF file; the remaining comments are provided below:
- I suggest insert in bracket, in the header of the Table A1 after the word “with” (EOB) and after the word “without” (Control)
- In table A2, in the appendix section, in the "Molds and yeasts" row at 7 days of maturation the Control and EBO values ​​are inverse compared to 20 and 45 days.
Is this a data entry error? Otherwise, how do you explain this behavior?
- I suggest insert in bracket, in the header of the Table 2 after the words “cows fed” (EOB) and after the words “or not” (Control).
- Please improve the resolution of the image in Figure 1, it is too blurry and the writing are not well defined.
- The description of the essential oil blend composition—i.e., 'Essential oil blend composed of eucalyptus oil (157.9 g/L), peppermint oil (32 g/L), and menthol crystals (55 g/L) (BronchoVest, Biochem, Germany)'—is already reported in the Materials and Methods section; I believe it is redundant to repeat it in every table and figure.

Author Response
Please see the attachment.
For research article
Response to Reviewer X Comments |
||
1. Summary |
|
|
Thank you very much for taking the time to review this manuscript. Please find the detailed responses below and the corresponding revisions/corrections highlighted/in track changes in the re-submitted files. |
||
2. Questions for General Evaluation |
Reviewer’s Evaluation |
Response and Revisions |
We appreciate the time and effort dedicated to reviewing this manuscript. The Materials and Methods section has been enhanced, and additional information regarding the research design has been included. The presentation of the results has also been improved, particularly in the tables. 3. Point-by-point response to Comments and Suggestions for Authors |
||
Comments 1: I suggest insert in bracket, in the header of the Table A1 after the word “with” (EOB) and after the word “without” (Control) Response 1: Thank you for your suggestion. We have made the requested change and inserted “(EOB)” after “with” and “(Control)” after “without” in the header of Table A1. |
||
Comments 2: In table A2, in the appendix section, in the "Molds and yeasts" row at 7 days of maturation the Control and EBO values ​​are inverse compared to 20 and 45 days. Is this a data entry error? Otherwise, how do you explain this behavior? Response 2: Thank you for pointing this out. You are correct — the values for "Molds and yeasts" at 7 days of maturation in Table A2 were mistakenly inverted between the Control and EOB groups. This was a data entry error, and we have corrected it in the revised version of the table. We appreciate your careful review. Comments 3: I suggest insert in bracket, in the header of the Table 2 after the words “cows fed” (EOB) and after the words “or not” (Control). Response 3: Thank you for your suggestion. As requested, we have inserted “(EOB)” after “cows fed” and “(Control)” after “or not” in the header of Table 2. Comments 4: Please improve the resolution of the image in Figure 1, it is too blurry and the writing are not well defined. Response 4: We thank the reviewer for the suggestion. We have replaced Figure 1 with a higher-resolution version to improve clarity and ensure that all text and details are clearly visible. Comments 5: The description of the essential oil blend composition—i.e., 'Essential oil blend composed of eucalyptus oil (157.9 g/L), peppermint oil (32 g/L), and menthol crystals (55 g/L) (BronchoVest, Biochem, Germany)'—is already reported in the Materials and Methods section; I believe it is redundant to repeat it in every table and figure. Response 5: We thank the reviewer for the observation. As suggested, we have removed the redundant information regarding the essential oil blend composition from the tables and figures where it was unnecessarily repeated. |

Round 2
Reviewer 3 Report
Comments and Suggestions for Authors
The article has been greatly improved, it is recommended to accept it